# HUMAN AND DEEP NEURAL NETWORK ALIGNMENT IN NAVIGATIONAL AFFORDANCE PERCEPTION

**Clemens G. Bartnik**
Informatics Institute
University of Amsterdam
Amsterdam, The Netherlands
c.g.bartnik@uva.nl

**Iris I.A. Groen**
Informatics Institute
University of Amsterdam
Amsterdam, The Netherlands
i.i.a.groen@uva.nl

## ABSTRACT

Moving through the world requires extracting navigational affordances from the immediate visual environment. How do humans compute this information from visual inputs? Over the last decade, Deep Neural Networks (DNNs) trained on visual recognition tasks have been shown to predict human perception remarkably well in the domain of object recognition, but their alignment with humans in other visual task contexts, such as spatial navigation, remain less understood. Here, we investigated the alignment of DNNs with human-perceived navigational affordances in a broad variety of visual environments by using explainable AI and different model training objectives. We curated a diverse set of naturalistic real-world indoor, outdoor man-made, and outdoor natural scenes. For each scene, we gathered human annotations identifying the objects present and collected drawings of path trajectories that participants would take through each scene. Quantitative analysis of the path annotations highlights that participants perceive and choose similar paths in each environment and thus diagnostic features for navigational affordances are present in the images. Using representational similarity analysis, we discovered that DNN features exhibit low correlations with information relevant to navigational affordance, such as mean pathways and floor segmentation. They demonstrate slightly better correlations with estimated depth information. However, these correlations are substantially lower than with the representational space of the contained objects. These findings illustrate that DNNs represent object information rather than representations of navigational affordances. This highlights that our path annotations are a rich and challenging benchmark to study human-DNN alignment and that current commonly used DNNs are yet not capturing navigational affordance representations well.

## 1 INTRODUCTION

Humans effortlessly perceive their environment as a coherent whole at a remarkable speed (Potter, 1975). Our perception extends beyond mere object recognition, encompassing a rich understanding of the surrounding scene which facilitates a variety of behavioral goals. Human psychophysical research over the years, including studies by Biederman et al. (1982) and Bar & Ullman (1996), has shown that scene perception involves both local properties, such as objects and their relationships, and global properties, including fixed immovable elements such as walls and sky. Perception of global properties is evident from humans' ability to infer scene meaning from unspecified blobs in specific spatial arrangements (Schyns & Oliva, 1994). Such perception of the 'gist' of a scene seems to be driven by visual properties like openness, depth, and navigability that are perceived before or in parallel with object information (Greene & Oliva, 2009). These findings inspired influential early computational models of object and scene perception such as HMAX (Poggio & Serre, 2013) and the GIST model (Oliva & Torralba, 2001).

While the representations of these models are fully understandable as they are constructed from 'handcrafted' features, inspired by hallmarks of the visual cortex, DNNs greatly outperform them in computer vision benchmarks (Krizhevsky et al., 2012; Kietzmann et al., 2019). Using supervised or unsupervised training regimes on large data sets DNNs learn rich feature representations optimized

for the task at hand. While deep learning models now achieve human-level performance at many visual tasks, it is not a necessity that their learned representations should resemble those of humans. Therefore, a new field of research has evolved studying potential determinants of representational alignment between humans and machine learning systems (Sucholutsky et al., 2023). So far, most studies focused on alignment of human behavior in the context of object or scene categorization tasks (Muttenthaler et al., 2023; Peterson et al., 2017; King et al., 2019; Groen et al., 2018).

Here, we delve into the alignment of humans and DNNs in a different domain: human navigational affordance perception, aiming to identify visual features that are important for spatial navigation. To achieve this, we collected human path drawings on a large set of 231 diverse scene pictures as well as ratings of contained objects. By comparing these representations with those of DNNs trained with different task objectives and with explainable AI (XAI) extracted representations, we show that commonly used networks align well with the annotations of contained objects but fail to capture navigational affordance perception. Specifically, our contributions are as follows:

- We collected path drawings for a diverse set of scenes containing indoor, outdoor natural and outdoor man-made environments. Quantitative analysis using Fréchet Distance highlights that humans consistently annotate possible pathways and thus that diagnostic features for navigational affordances are detectable.

- We confirm findings from previous research indicating that DNNs are well aligned with object ratings. However, we also demonstrate that the representational space of average path trajectories is not well captured by commonly used DNNs including various architectures, training sets, and task objectives.

- We test other DNN-derived feature importance visualization maps and show that explicit spatial representation of potentially navigational relevant information, such as floor segmentations, are not well captured, but that estimated depth information is better captured by the set of DNNs we tested here.

Overall, our data provide a new, rich, and interesting benchmark that captures human navigational affordances perception, but it is not yet well aligned with commonly used DNN representations.

## 2 RELATED WORK

About a decade ago, DNNs started to reach human-level performances in object recognition (Krizhevsky et al., 2012) whilst also outperforming shallow computer vision models in predicting neural responses in intermediate and higher areas of the ventral stream of the visual cortex (Cadieu et al., 2014; Khaligh-Razavi & Kriegeskorte, 2014) across various modalities including electrophysiology (Cadieu et al., 2014), fMRI (Guclu & van Gerven, 2015), MEG (Cichy et al., 2016), and EEG (Greene & Hansen, 2018) making them promising computational models to understand the human visual system (see e.g., Kietzmann et al., 2019; Storrs & Kriegeskorte, 2019; Serre, 2019, for reviews). Internal representations of DNNs also have been aligned with human perceptual assessments of objects, revealing that they mirror the perception of object similarity observed in both monkeys (Rajalingham et al., 2018) and humans (Kubilius et al., 2016). While most prior research focused on object recognition, DNNs also exhibit accurate predictions of neural responses during scene (Cichy et al., 2017; Greene & Hansen, 2018; King et al., 2019) and action recognition (Güçlü & Gerven, 2017). However, these studies primarily focused on basic object or scene categorization tasks.

How can we incorporate more challenging behavior, specifically spatial navigation behavior, in the study of alignment of DNNs and human representations, in order to better understand the visual system and improve computational models? Bonner & Epstein (2017) introduced path drawings of indoor environments as a suitable representation to uncover where in the brain navigational affordances are processed. Follow-up work by Bonner & Epstein (2018) showed that mid-level layer activations of a scene-trained AlexNet show the highest alignment with navigational affordance representations, and determined that high spatial frequencies and cardinal orientations in the lower visual field are mainly driving these activations. Dwivedi et al. (2021) furthermore showed that representations of scene parsing networks better explain the same set of brain activity recordings in scene-selective regions over scene classification-trained models.

## 3 Methods

### 3.1 Stimuli

We collected a set of 231 high-resolution color photographs (resolution: 1024×1024 pixels) from Flickr, to obtain a novel set of images that are not part of any commonly used large-scale image database (e.g. ImageNet or Places). Images depicted typical everyday environments devoid of prominent objects, humans, or animals and were captured from a human-scale, eye-level perspective (see Supplementary Figure 5 for examples). Diverging from previous research that predominantly utilized indoor imagery (e.g., Bonner & Epstein, 2017; Zamir et al., 2018), our study uses scenes from three distinct types of environments (indoor, outdoor natural, and outdoor man-made) a common taxonomy used in scene classification datasets (Zhou et al., 2018).

### 3.2 Collecting behavioral ratings

#### 3.2.1 Procedure

We conducted an online experiment designed using the Gorilla Experiment Builder (Anwyl-Irvine et al., 2020) to quantify navigational affordances of our stimuli set gathering possible path trajectories similar to the paradigm used by (Bonner & Epstein, 2017). Initially, 167 participants were recruited from prolific.ac (Palan & Schitter, 2018), of whom 81 were female. 15 participants did not complete the study, resulting in a final count of 152 participants. The participants had a median age of 25 years old (range 18 - 74 years, $SD$ = 14). We used stringent selection criteria provided by Prolific to screen for reliable participants. All participants had normal or corrected-to-normal vision. Participants were compensated with 7.5 £ . The experiment began with an introductory screen that informed participants about the experiment, and a screen where they gave informed consent to participate in our study. The study was approved by the ethical committee of our institution.

#### 3.2.2 Object ratings

To gather annotations of the presence of certain objects in the scene we had the participants view each of our 231 images for 1 second followed by a response screen with six button options (Building/Wall, Tree/Plant, Road/Street, Furniture, Body of water, Rocks/Stones). This list of response options was inspired by previous research Fei-Fei et al. (2007) characterizing objects in scenes.

#### 3.2.3 Task description Path drawing

During the task, each trial consisted of one of our 231 images with a yellow overlay (25 x 1024 pixel) at the bottom indicating where participants should start drawing their path. Participants used their computer mouse to draw a red line indicating a path they would use to move through the depicted scene (see 1 **A** for an example drawing). Each participant indicated one possible pathway. As Gorilla only provides JPEG images of the path annotations, paths were manually redrawn in a self-written tool in Python. The paths were saved as a CSV file containing sequences of x, and y coordinates in the image space. For each image, all path sequences were resampled to match the same length of points as the longest trajectory. These were resampled to have the same length.

With ($M$ = 21.56, $SD$ = 0.97) path annotations per image, we obtained a set of trajectories $\{T_1, T_2, \ldots, T_m\}$. Each trajectory is represented as a sequence of points $\{(x(t_i), y(t_i))\}$ in the 2D image space, capturing the x and y coordinates at discrete time intervals $t_i$. These trajectories enabled us to create heatmaps by averaging the 2D path annotations for each image 1 **A**. As each participant annotated only one path, this setup provided an opportunity to examine the variability and similarity between the individual paths.

#### 3.2.4 Quantifying navigational affordances

To perform a quantitative comparison of these trajectories, we utilized the discrete Fréchet Distance (Eiter & Mannila, 1994). For two trajectories $T_a$ and $T_b$, the Fréchet Distance $F(T_a, T_b)$ is expressed as:

$$F(T_a, T_b) = \inf_{\alpha, \beta} \max_{t \in [0,1]} \{ dt(T_a(\alpha(t)), T_b(\beta(t))) \tag{1}$$

where $\alpha$ and $\beta$ are continuous non-decreasing functions mapping the interval $[0, 1]$ to the domains of $T_a$ and $T_b$, respectively, and $d$ represents the Euclidean distance. This metric is intuitively understood as the minimum leash length necessary for a person to walk a dog along two separate paths (the trajectories) from start to end without backtracking.

Subsequently, we computed the average Fréchet Distance across all possible pairs of trajectories for each image, providing insights into the degree of agreement regarding the most apparent and natural pathway through each scene. To put the mean Fréchet Distances into perspective we also created a distribution of mean Fréchet Distance representing randomly assigned path trajectories. This involved randomly allocating 20 pathways to each image from the entire collection of path annotations gathered during our online experiment. Subsequently, we calculated the pairwise Fréchet Distances using the previously described method. To test if the random pathways exhibit a higher average Fréchet Distance, we conducted a t-test to determine if the two distributions of average average Fréchet Distances are significantly different from each other.

There are various ways to transform path annotations into a navigational affordance feature space. Bonner & Epstein (2017) adopted an approach where they classified their path annotations based on whether they appeared on the left, center, or right side of the image. Here we utilized a different approach dividing each image into segments of 20x20 pixels and averaging the path information within each segment. This method offers the advantage of a more detailed representation where fine grained spatial information of navigational affordances are preserved. Additionally, with this approach alignment can be computed with any form of heatmap (e.g. XAI depictions or other visual hypotheses) by simply vectorizing the image.

## 3.3 Vision Models

We considered a diverse set of commonly used pre-trained neural networks with different architectures, trained on different datasets. We used ResNet50 (He et al., 2015), AlexNet (Krizhevsky et al., 2012), VGG16 (Simonyan & Zisserman, 2014) as vanilla CNN architectures trained on ImageNet-1K (Deng et al., 2009). To examine a possible influence of the dataset used for training the network, we also included a ResNet50 trained for scene classification on Places365 (Zhou et al., 2018) and a R50-FPN (Wu et al., 2019) trained for Scene parsing on ADE20k (Zhou et al., 2017), as well as two action recognition models from PySlowFast (Fan et al., 2020) with X3D m and SlowFast using a ResNet101 backbone. Lastly, we used three CLIP models from OpenCLIP (Radford et al., 2021) with ResNet101, VIT-B-16, and VIT-B-32 backbones. Feature activations of our 231 images were extracted from the pre-trained networks using the Net2Brain toolbox (Bersch et al., 2022). For each network, we extracted activations from the Net2Brain predefined layers. Extracted features were standardized by removing the mean and scaling to unit variance.

## 3.4 Image derived visual feature visualizations

To better understand which visual information is important for navigational affordances we explored various DNN-derived feature spaces that we hypothesize to contain navigational affordance related information.

### 3.4.1 Floor Segmentation

An intuitive approach to identifying navigational affordances involves focusing on the ground plane of a scene, as it naturally serves as the foundation for pathways. Unlike the method proposed by Bonner & Epstein (2017), in which participants indicate all possible pathways that they perceive which estimates the full navigable area, our single path annotations focus on the most likely path therefore being only a subregion in the floor segmentation. To estimate the ground plane in our images, we used a Cascade Segmentation Module trained on the ADE20k dataset (Zhou et al., 2018), which segments various elements of the scene. Subsequently, we isolated and generated binary segmentation maps for labels indicative of the ground plane. These maps were then used to calculate pairwise correlations, serving as the basis for our RDMs in this representational space.

### 3.4.2 MONOCULAR DEPTH ESTIMATION

To navigate through an environment it's important to perceive open unobstructed areas. Therefore depth information might be an important driver for navigational affordance perception. To test this we utilized a state-of-the-art monocular depth estimation model (Miangoleh et al., 2021). This model generates high-resolution depth maps from single RGB images by first producing depth estimations at multiple resolutions. Afterwards merging these into a single high-resolution depth map considering content-specific details and discrepancies across different resolutions. As described above we computed the pairwise correlation between those depth maps across our images resulting in an RDM representing which scenes are dissimilar to each other in regards to depth information.

### 3.4.3 LAYER-WISE RELEVANCE PROPAGATION

Bonner & Epstein (2018) showed that a scene-trained AlexNet represents navigational affordance representations likely by encoding the presence of high spatial frequencies, rectilinear features, and cardinal orientations. A popular explainable AI used to visualize neural network decisions is Layer-wise Relevance Propagation (LRP; Bach et al., 2015). This method backpropagates the prediction output through the layers in reverse order, attributing relevance scores to each input pixel. This process results in a fine-grained heat map highlighting the most influential pixels in the input image. As displayed in 4 these maps highlight edges and high spatial frequencies, making this a promising feature representation to test the importance of these features further. We passed our images through a VGG16 trained on Places 365 (Zhou et al., 2018) and created LRP heatmaps for each image using the investigate toolbox (Alber et al., 2019) and created RDMs using pairwise correlations of LRP heatmaps.

### 3.4.4 GRAD-CAM

As salient objects in a scene could inform navigational affordances we utilized Gradient-weighted Class Activation Mapping (Grad-CAM; Selvaraju et al., 2020) to visualize the areas in the input image that are most important for the predictions for the respective class. We used a VGG16 trained on Places365 for scene classification (Zhou et al., 2018) and created Grad-CAM heat maps for the third layer or the convolution block five, the last feature extraction block. The pairwise correlations of the resulting heatmaps again formed the RDM for this feature space.

### 3.5 REPRESENTATIONAL SIMILARITY ANALYSIS

We utilized representational similarity analysis (RSA; Kriegeskorte, 2008) to measure the correlation between human path annotations and feature activations of neural networks and DNN-derived visual feature visualizations. For each representational space, we created representational dissimilarity matrices (RDMs) based on pairwise 1-Pearson correlation. For both the path annotations and the visual features derived from images, we transformed the averages of the 20x20 image segments into vector form to serve as feature vectors. Regarding the vision models, we utilized the standardized activations from each layer as feature vectors. For the object representational space we computed RDMs using pairwise Euclidean correlation distances from the proportion of participants that annotated the presence or possibility of a given label (e.g., proportion of participants that annotated the scene as containing a road). The Euclidean distance metric was chosen due to the sparsity of the annotation options. Alignment was measured through Spearman's rank correlation coefficient between the RDMs.

## 4 RESULTS

We used path drawings to study human and machine learning systems alignment in the domain of navigational affordance perception. We aim to understand what visual features underlie human visual processing for spatial navigation and to what extent DNNs capture these features. First, we present our path annotation data and determine their reliability and consistency. Then, we show how well a variety of different DNNs with varying architecture and training datasets capture navigational affordances in the form of path drawings and object representations via annotations of contained objects. Subsequently, we explore the connection between different 2D spatial DNN-derived visual

features, such as segmentation network outputs or estimated depth maps, and explainable AI feature importance maps (LRP and Grad-CAM), examining their alignment with navigational affordances. Finally, we delve deeper into how the DNN feature activations relate to these individual visual feature representations.

## 4.1 PATH ANNOTATIONS IN COMPLEX ENVIRONMENTS ARE CONSISTENT ACROSS PARTICIPANTS

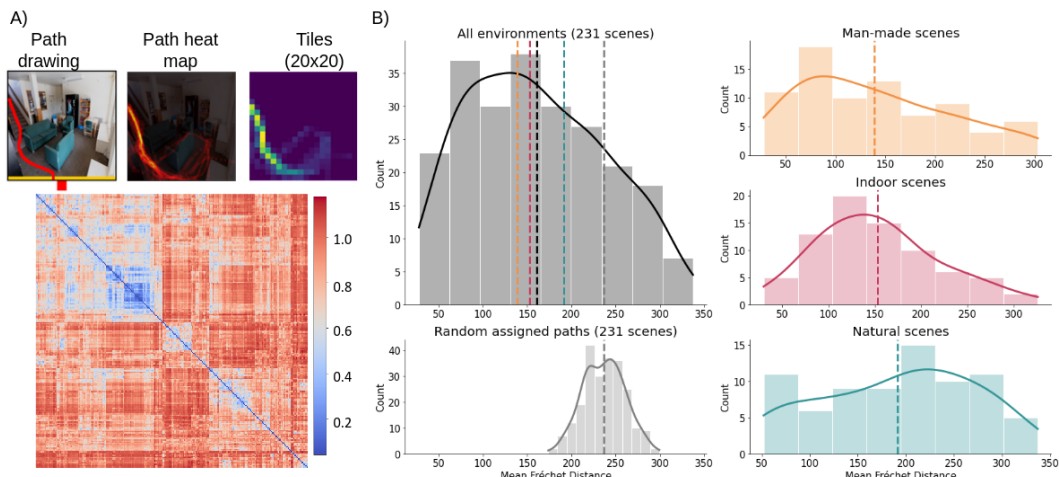

Figure 1: Capturing human navigational affordances: (A) illustrates the process of deriving an RDM from path annotations made by participants across various scenes. Path drawings were first averaged into a single heat map per scene, then split into 20x20 tiles representing the average path information in each tile (top). Subsequently, all tiles were flattened and then pairwise correlated using a 1-Pearson correlation, yielding a pairwise distance matrix reflecting the degree of overlap in navigational affordance locations between different images (bottom). (B) depicts the distribution of mean Fréchet Distances across all 231 images (left) or split by their environment label (right). Below is a distribution of mean Fréchet Distances for randomly assigned pathways. Overall we can see that the path annotations are consistent as the mean value for randomly assigned paths is substantially higher.

By using the Fréchet Distance (Eiter & Mannila, 1994) we measured how well participants agreed on the most likely path to traverse a given scene. The left panel of 1 **B** depicts a histogram of mean Fréchet Distances across our full 231 image dataset. Participants tended to agree on where to move through the scene which is reflected by the overall mean Fréchet Distances score ($M = 167.73$, $SD = 77.63$) which is significantly lower [$t(460) = 14.38$, $p < 0.001$] compared to the mean Fréchet Distances when randomizing paths across images ($M = 236.88$, $SD = 23.43$) (see 1 **B**). We also computed the path consistency for each environment type separately and found the lowest mean Fréchet Distances in outdoor man-made environments ($M = 150.67$, $SD = 74.1$), closely followed by indoor scenes ($M = 158.23$, $SD = 71.65$). These environments show no significant difference [$t(150) = 1.19$, $p = 0.23$] to each other suggesting that these types of environments have well-defined pathways that participants would choose to navigate. In natural scenes ($M = 194.27$, $SD = 79.81$) we found the highest mean Fréchet Distance which was significantly different from man-made environments [$t(150) = 4.21$, $p < 0.001$] but not significantly different from indoor environments [$t(150) = 3.28$, $p = 0.0013$], and still significantly lower than randomized paths [$t(150) = 7.64$, $p < 0.001$]. This highlights that in natural scenes on average more diverse possible pathways are perceived. However, paths are still consistently drawn in the ground plane even if somewhat more dispersed than in man-made environments.

Overall, this quantification of path annotations shows that across multiple types of environments, participants are highly consistent in annotating possible pathways, suggesting that diagnostic visual features for spatial navigation are present in the images. This makes these annotations an intriguing space to study human and machine alignment in the context of navigational affordance perception.

## 4.2 ALIGNMENT OF DNNS WITH MEAN PATH AND CONTAINED OBJECT ANNOTATIONS

Figure 2: Comparison of DNN models best-correlating layer with mean path RDM, and object ratings in the online experiment using Spearman's rho correlation coefficient.

Next, we evaluated how well these navigational affordances in the form of path annotations are captured by feature activations of common DNN models. Hence, we used a variety of pre-trained DNNs covering different architectures, training datasets, and training regimes (see Methods). The correlations between path annotation-derived RDMs and these networks are depicted in 4.2. While the DNNs capture some information about navigational affordances overall correlations are very low. We observe the highest correction of the mean paths with a ResNet50 trained on ImageNet ($rho$ = 0.08, $p$ <0.001) and CLIP ViT-B-16 ($rho$ = 0.08, $p$ <0.001) trained on the OpenCLIP dataset. Different training sets like Places365, ADE20k, or Kinect700 seem to not improve the DNN alignment with path annotations.

In contrast, we observe that the feature activations of the DNNs correlate highly with annotations of the objects present in the scenes. Here we find the highest correlation with the CLIP ViT-B-32 ($rho$ = 0.7, $p$ <0.001) model. Interestingly the ResNet50 model trained on ImageNet exhibits the lowest correlations ($rho$ = 0.26, $p$ <0.001). Additionally, we can observe that ResNet50s are better aligned with the object representation when trained on alternative datasets like Places365 ($rho$ = 0.65, $p$ <0.001) or for scene parsing on ADE20k ($rho$ = 0.66, $p$ <0.001).

These results confirm prior research that the feature activations of DNNs are well aligned with human behavior regarding objects contained in scenes, but show that they fail to capture mean path annotations.

## 4.3 COMPARATIVE ANALYSIS OF FEATURE SPACES IN SCENE REPRESENTATION

As we showed in the previous section feature activations of DNNs do not capture navigational affordance well. But what kind of information might be important for navigational affordances and are these better represented by DNNs? To explore this question we compute the interrelations between different DNN-derived visual feature importance maps for all our 231 images 4.3 **B**. We find the highest correlation between the mean path annotations and floor segmentation representations ($rho$ = 0.24, $p$ <0.001), which is intuitive as pathways need to be on the ground plane. Still, our path annotations rather highlight a preferred pathway, and not an annotation of the full navigable surface

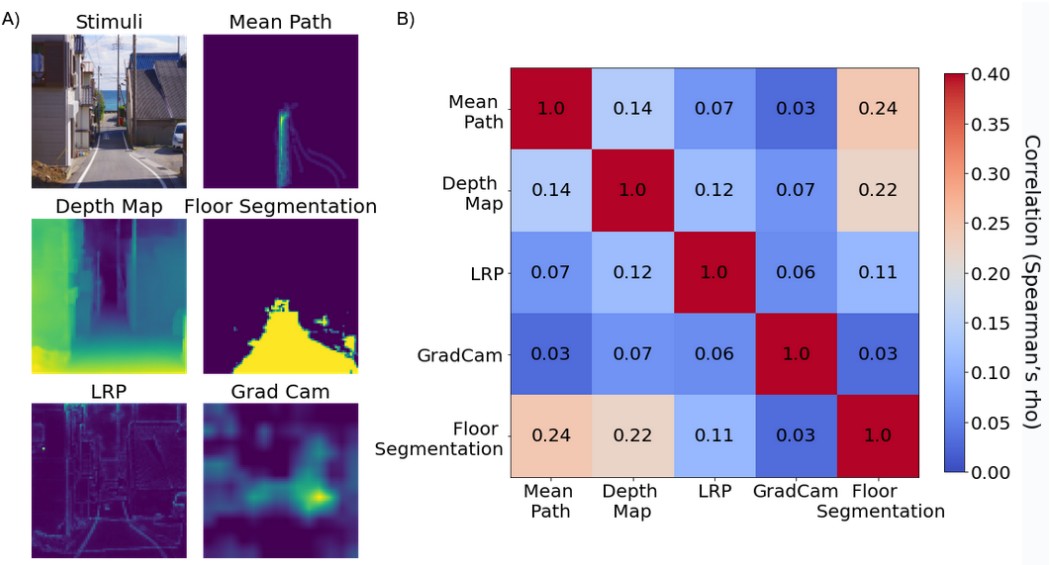

Figure 3: (A) Examples of DNN-derived spatial importance maps. The initial row presents stimuli and the average path annotation. The subsequent row showcases monocular depth estimation alongside floor segmentation maps extracted using ADE20k. The final row depicts spatial importance maps from explainable AI techniques, LRP and Grad-CAM, applied to VGG16 trained for scene classification on Places365. (B) Correlation matrix comparing DNN-derived feature importance maps with mean path annotations and with one another using Spearman's rho. The matrix displays the inter-relatedness of the various spaces (blue for lower, red for higher correlations).

area, hence the correlation is still relatively modest. The mean path also shows a notable correlation with the depth map ($rho = 0.14$, $p < 0.001$), suggesting that the perceived path trajectory is somewhat aligned with depth cues in the scene. However, we find a comparatively higher correlation between floor segmentation and depth estimation representations ($rho = 0.22$, $p < 0.001$). Besides using the 'classic' way of aligning DNNs with human behavior through feature activations we can utilize other visual hypotheses to test DNN alignment. Here we use LRP to capture high spatial frequency information ($rho = 0.07$, $p < 0.001$) and Grad-CAM ($rho = 0.03$, $p < 0.001$). The low Grad-CAM correlation shows that the areas in the image that are most decisive for classification, in this case, the scene class, are not part of the most preferred pathways.

Overall, these correlations reveal which feature spaces are associated with navigational affordances, especially floor segmentation and depth estimation, and enable us to test how well DNNs capture these visual feature hypotheses.

Figure 4 illustrates how well DNNs capture these other visual feature spaces in comparison to the mean path annotations. DNNs demonstrate marginally higher correlations with the floor segmentation representation compared to the mean path representation. Here we find that CLIP R101 ($rho = 0.11$, $p < 0.001$) correlates highest. This showcases once more that DNNs don't capture navigational affordance related representations well. While higher correlations are observed between DNN layer activations and the estimated depth map representation, they remain substantially lower than those for object information. The VGG16 model trained on ImageNet exhibits the highest correlation ($rho = 0.24$, $p < 0.001$) among them. This suggests that the representations captured by the DNNs are more closely associated with depth information, as opposed to navigational affordance features.

Together, these results showcase that common DNNs are not effective at capturing navigational affordances, such as mean pathways and related features like floor segmentation, and instead represent object information. Nonetheless, our approach enables a deeper analysis of how well certain features align with DNN representations, distinguishing between those that are captured and those that are not

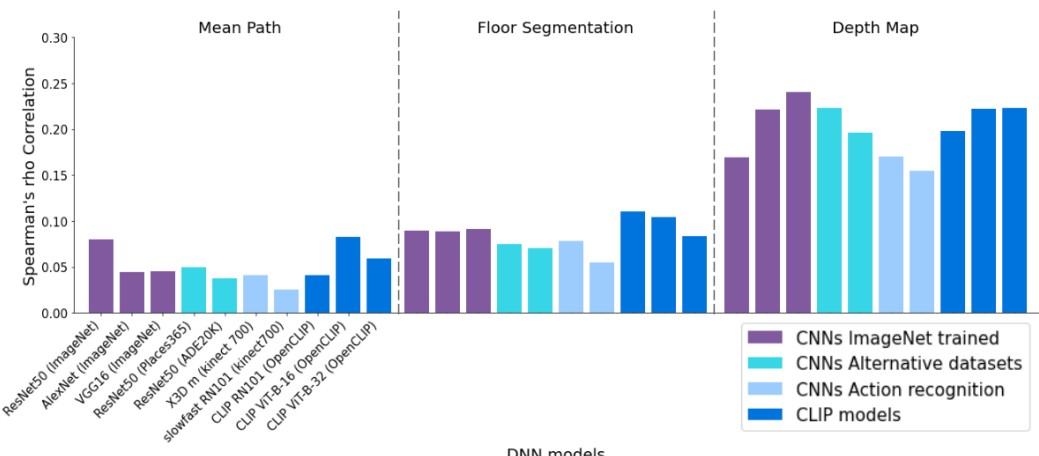

Figure 4: Comparison of DNN models best-correlating layer with Comparison mean path, floor segmentation, and estimated depth map RDMs.

## 5 DISCUSSION

In this study, we investigate how well DNNs align with human spatial navigation, specifically in perceiving navigational affordances through path drawings. Our findings reveal a striking discrepancy: although participants consistently annotated pathways, these navigational representations are poorly captured by feature activations in DNNs. Instead, DNNs primarily represent object representations. Further analysis using 2D spatial visual features derived from DNNs further demonstrates their inadequacy in representing other visual features related to navigation, such as floor segmentation.

Using the Fréchet Distance as a quantitative measure, we show that humans annotate scene images in our dataset with high consistency. However, while the Fréchet Distance provides a clear method for assessing path coherence, it sometimes underestimates how coherent the paths actually are. This is particularly evident in scenarios where a scene presents two distinct yet viable paths, leading to a higher average Fréchet Distance—surprisingly, sometimes even higher than that of randomized pathways (e.g., Supplementary Figure 6). Given these limitations, future research should explore alternative measures for evaluating path consistency. This will further enhance the human navigational affordance benchmark with which alignment can be computed.

Our tiling approach involves segmenting images into tiles to analyze feature representations, providing a more rigorous assessment of alignment by specifically looking for spatially specific alignment. This approach reveals that DNNs are better at recognizing depth information rather than navigational cues. To gain a comprehensive understanding of the features DNNs represent, future research should explore a wider range of visual hypotheses

To improve the alignment of DNNs with human navigational affordance perception, future studies could consider training models using mean path heatmaps or segmentation masks. However, such an approach would necessitate a significantly larger dataset of human-annotated paths than what we have collected. An alternative, less resource-demanding strategy might involve using mean path heatmaps to refine the focus of DNNs, as suggested by Fel et al. (2022). This method would adjust the networks' attention mechanisms to prioritize areas highlighted by human annotations. Ultimately, this could enable DNNs to develop representations that more accurately reflect the characteristics humans use to perceive navigational affordances within a scene.

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

## A  APPENDIX

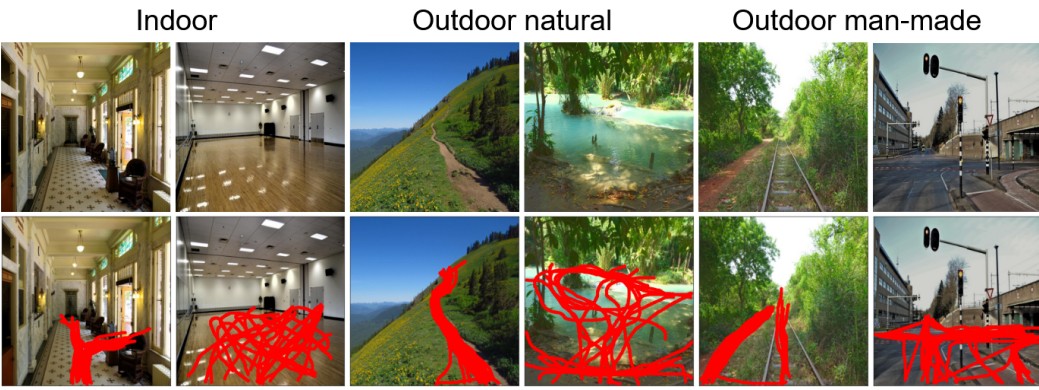

Figure 5: Example Stimuli for our three different types of environments and the respective path annotations from the online experiment. Each image was annotated by over 20 participants marking a single pathway they would use to navigate the depicted scene.

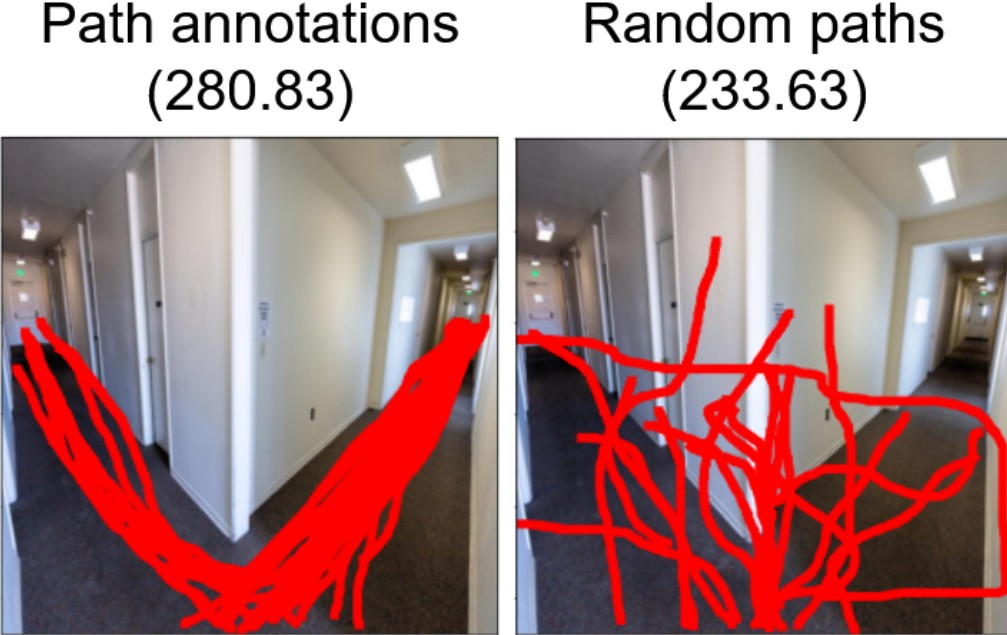

Figure 6: Fréchet Distance as a quantitative measure for path similarity underestimates the consistency of our path annotations. As there are two diverging path options the overall mean pairwise Fréchet distance between all paths is quite large (280.83) larger than for randomly placed path trajectories (233.63), yet we visually can observe that the trajectories are very similar between the participants.

