# OpenReview forum: "Human and Deep Neural Network Alignment in Navigational Affordance Perception"
_ICLR.cc/2024/Workshop/Re-Align — ICLR 2024 Workshop Re-Align Poster_

### Official Review · Reviewer_4ybZ · 2024-02-23
**Interesting research paper which could be improved with a bit more clarity.**

**Rating:** 2
**Fit:** 3
**Confidence:** 2

**Workshop Review:**

***Summary***

The paper collected a human annotated dataset of path trajectories which indicate how the participant would navigate the respective scene. These annotations are then correlated with the activations of different neural networks. Further the path maps are correlated with heatmaps from depth estimation, LRP, GRAD-CAM, depth estimation and floor segmentation. The authors find that the networks exhibit high correlation with the presented objects in the scene but low correlations with information relevant to navigational affordance.

***Strengths***

* The idea is interesting and novel.
* Different methods are considered in the correlation analysis (depth estimation, LRP, GRAD-CAM, depth estimation, floor segmentation)
The result that there is not high disagreement between different human path annotations is interesting!

***Weaknesses***
* It is unclear to me which values of the output were explained by LRP. This could make a big difference.
* While dense models (segmentation, object detection) have been considered, it would be interesting to consider the representations of diffusion models in the analysis.

***Conclusion***

The idea is interesting and the authors explored different ways of correlating path trajectories with deep neural networks. The clarity of the paper could be improved though. Overall, I would recommend acceptance.

**Reason For Not Giving Higher Score:**

The paper is lacking clarity in some paragraphs (for example what exactly is explained with LRP). Further, it is in some way expectable that models that were not trained to find a path are also not correlated with path annotations.

**Reason For Not Giving Lower Score:**

I find the idea and dataset interesting and it could help to spark further research in the community. The paper raised an interesting question and provided reasonable experiments to answer that question.

**Reviewer Domain:**

machine learning

---

### Official Review · Reviewer_rhTF · 2024-02-25
**DNNs don't exhibit human-like navigational affordance properties**

**Rating:** 2
**Fit:** 3
**Confidence:** 3

**Workshop Review:**

This is a fairly straightforward study that aims to compare navigational affordance behavior in humans with the internal representations of deep networks. The authors collect scenes and human annotated objects and scene-path annotations. Using RSA, they find that DNN features have low correlation with scene annotations (compared with object annotations). The authors then conclude that DNNs represent objects rather than scene information.

This is an interesting idea and a promising approach but the study is overly simplistic in both its implementation and logic.

The reason I say this is because the comparison here is between model "internals" and human "output"/behavior. A rephrasing of the inference from these results is that scene-path representations are not explicitly represented in model units (like object representations). However there may still be ways to read out this information from basic scene primitives. It is quite possible that human read-out mechanisms utilize such primitives for a complex behavior like scene path detection. I would urge the authors therefore to build on this work and explore variants of this approach and move beyond RSA in future work.

**Reason For Not Giving Higher Score:**

This work seems to be preliminary data and hasn't yet explored more complex versions of their hypothesis. I want to encourage the authors but it isn't yet a talk.

**Reason For Not Giving Lower Score:**

This work seems to be preliminary data and hasn't yet explored more complex versions of their hypothesis. I want to encourage the authors but it isn't yet a talk.

**Reviewer Domain:**

cognitive science

---

### Official Review · Reviewer_wUAb · 2024-02-26
**Interesting angle on affordances to study network  and human alignment in perception**

**Rating:** 2
**Fit:** 3
**Confidence:** 3

**Workshop Review:**

The paper studies how well deep neural networks capture affordances for navigation.  They collected a set 231 high-resolution color photographs and ask participants online to draw path trajectories given an initial point. In order to measure the agreement of the trajectories between participants,  they used the Fréchet distance. Subsequently,  they turn into deep neural networks, trained for classification, depth and segmentation to study which networks align better with human affordances.

I think the paper offers a new point of view that has not been studied on neural networks. I also appreciate the new dataset collected that hopefully will be made available to keep advancing on this topic. As well I appreciate the authors delineating the current limitations of the work.

**Reason For Not Giving Higher Score:**

I think the approach is great and have a potential  to become an interesting research agenda. I think the scale still is too small though to make it more impactful, but I think is a great first step.

**Reason For Not Giving Lower Score:**

The paper offers a new view on the field and shows that there is potential in doing research on this particular field.

**Reviewer Domain:**

cognitive science

---

### Decision · Program_Chairs · 2024-03-02

Accept (Poster)